# Mechanical forces drive a reorientation cascade leading to biofilm self-patterning

Japinder Nijjer[1], Changhao Li [2], Qiuting Zhang [1], Haoran Lu[1], Sulin Zhang [2,3✉] & Jing Yan [1,4✉]

In growing active matter systems, a large collection of engineered or living autonomous units metabolize free energy and create order at different length scales as they proliferate and migrate collectively. One such example is bacterial biofilms, surface-attached aggregates of bacterial cells embedded in an extracellular matrix that can exhibit community-scale orientational order. However, how bacterial growth coordinates with cell-surface interactions to create distinctive, long-range order during biofilm development remains elusive. Here we report a collective cell reorientation cascade in growing *Vibrio cholerae* biofilms that leads to a differentially ordered, spatiotemporally coupled core-rim structure reminiscent of a blooming aster. Cell verticalization in the core leads to a pattern of differential growth that drives radial alignment of the cells in the rim, while the growing rim generates compressive stresses that expand the verticalized core. Such self-patterning disappears in nonadherent mutants but can be restored through opto-manipulation of growth. Agent-based simulations and two-phase active nematic modeling jointly reveal the strong interdependence of the driving forces underlying the differential ordering. Our findings offer insight into the developmental processes that shape bacterial communities and provide ways to engineer phenotypes and functions in living active matter.

[1] Department of Molecular, Cellular and Developmental Biology, Yale University, New Haven, CT, USA. [2] Department of Engineering Science and Mechanics, Pennsylvania State University, University Park, PA, USA. [3] Department of Biomedical Engineering, Pennsylvania State University, University Park, PA, USA. [4] Quantitative Biology Institute, Yale University, New Haven, CT, USA. ✉email: suz10@psu.edu; jing.yan@yale.edu

The spatiotemporal patterning of cells is a fundamental morphogenetic process that has profound effects on the phenotypes and functions of multicellular organisms[1–3]. In the prokaryotic domain, bacteria are often observed to form organized multicellular communities surrounded by extracellular matrices[4,5], known as biofilms[6,7], which are detrimental due to persistent infections, clogging of flows, and surface fouling, but can be beneficial in the context of wastewater treatment[8] and microbial fuel cells[9]. During development, biofilms exhibit macroscopic morphological features ranging from wrinkles, blisters, to folds[10–12]. At the cellular scale, recent progress in single-cell imaging has revealed the reproducible three-dimensional (3D) architecture and developmental dynamics of biofilms[13–16]. However, how cellular ordering emerges from individual bacterium trajectories remains poorly understood. In particular, it remains unclear how cell proliferation is coordinated with intercellular interactions to elicit robust self-patterning against bacteria's inherent tendency to grow in an unstructured manner[17–19]. An understanding of how individual cell growth links to collective patterning as a result of self-generated forces can provide insights into the developmental program of biofilms, their physical properties[20], and the engineering of living and nonliving active-matter analogs[21,22].

To bridge the gap between interactions at the cellular scale and patterns at the community scale, here we combine single-cell imaging and agent-based simulations to reveal the underlying mechanism for self-patterning in biofilm formed by *Vibrio cholerae*, the causal agent of the pandemic cholera[23]. We observe that biofilm-dwelling bacteria self-organize into an aster pattern, which emerges from a robust reorientation cascade, involving cell verticalization in the core and radial alignment in the growing rim. We reveal that the verticalized core generates a directional flow that drives radial alignment of the cells in the periphery, while the rim generates compressive stresses that expand the verticalized core, leading to a robust, inter-dependent differential orientational ordering. Based on these findings, we derive a two-phase active nematic model for biofilm self-patterning, which is potentially generalizable to other developmental systems with growth-induced flows[24,25]. Our findings suggest that the self-generated cellular force landscape, rather than chemical signaling or morphogen gradients as often seen in eukaryotic cells[26], control pattern formation in biofilms.

## Results

### *V. cholerae* biofilms self-organize into aster patterns.
We imaged the growth of *V. cholerae* biofilms confined between glass and an agarose gel at single-cell resolution (Fig. 1a). We used a constitutive biofilm producer locked in a high c-di-GMP state[27] and focused on the biophysical aspects of self-organization. To simplify our study, we focused on a mutant missing the cell-to-cell adhesion protein RbmA[15,28]—this strain is denoted as WT*—although our analysis is equally applicable to strains with cell-to-cell adhesion (Supplementary Fig. 1). Using confocal microscopy, the 3D architecture of the biofilms was captured over time from single founder cells to mature biofilms consisting of thousands of cells (Fig. 1b and Supplementary Video 1). An adaptive thresholding algorithm was used to segment individual cells in the 3D biofilm (Supplementary Fig. 2 and Supplementary Information Section 1) from which the location and direction of each rod-shaped bacterium were identified (Fig. 1c–f). Strikingly, cells in the basal layer of WT* biofilms reproducibly self-organized into an aster pattern, consisting of a core with tilted or verticalized cells and an outer rim with radially aligned cells (Fig. 1d and Supplementary Fig. 3). We found a similar pattern of verticalized cells surrounded by a rim of radially aligned cells when the biofilm was grown without an overlain agarose gel[13–15], albeit with a smaller degree of radial alignment (Supplementary Fig. 4).

To quantify the degree of cell ordering in the basal layer, we defined a radial order parameter[29] $S = 2\langle(\hat{\mathbf{n}}_{\parallel} \cdot \hat{\mathbf{r}})^2\rangle - 1$, where $\hat{\mathbf{n}}_{\parallel}$ is the projected cell direction on the basal plane ($\hat{\mathbf{n}}_{\parallel} = (\hat{\mathbf{n}} - (\hat{\mathbf{n}} \cdot \hat{\mathbf{z}})\hat{\mathbf{z}})/|(\hat{\mathbf{n}} - (\hat{\mathbf{n}} \cdot \hat{\mathbf{z}})\hat{\mathbf{z}})|$), $\hat{\mathbf{r}}$, $\hat{\mathbf{n}}$ and $\hat{\mathbf{z}}$ are the unit vectors along the radial, cell, and vertical directions, respectively, and the angled brackets indicate averaging over all cells (Fig. 1g). $S$ equals 1 for cells that are radially aligned, 0 for cells that are randomly oriented, and −1 for cells that are aligned in a vortex. We found that cells in WT* biofilms exhibited a reproducible tendency to align radially ($S = 0.54 \pm 0.07$). Since previous work has shown that cell-to-surface adhesion controls overall biofilm morphology[12,14,30], we hypothesized that cell-to-surface adhesion also mediates the dynamic core-rim patterning of biofilms. To test this hypothesis, we deleted the genes encoding cell-to-surface adhesion proteins Bap1 and RbmC[31–33] ($\Delta BC$) and found that the radial order was destroyed in the resulting biofilms and cells assumed random orientations in the basal plane with $S = 0.11 \pm 0.11$ (Fig. 1e, g). Concomitant with the disorder was the absence of a verticalized core; most cells in the basal layer were parallel to the substrate. We further confirmed the important role of cell-to-surface adhesion by titrating *rbmC* expression: increasing cell-to-surface adhesion enhanced the self-patterning, resulting in more verticalized cells and stronger radial alignment (Fig. 1h and Supplementary Fig. 5). Furthermore, removing the extracellular matrix by deleting the key *Vibrio* polysaccharide biogenesis gene *vpsL*[34,35] resulted in locally aligned microdomains of horizontal cells without long-range order ($S = 0.02 \pm 0.08$; Fig. 1f, g), in line with previous studies on growing two-dimensional (2D) bacterial colonies[18,19]. These observations suggest that exopolysaccharide production controls a local order-to-disorder transition, whereas cell-to-surface adhesion controls a global order-to-disorder transition.

To determine the driving forces behind the observed orientational ordering, we extended a previous agent-based model[36], taking into account cell-to-cell and cell-to-surface interactions (Supplementary Information Section 2 and Supplementary Fig. 6). Our agent-based modeling reproduced the observed aster pattern formation in adherent cells but not in nonadherent cells, in agreement with experiments (Supplementary Fig. 7 and Supplementary Video 2). As the agent-based model only incorporates mechanical interactions without any biochemical signals, our results suggest that the emergent patterns originate primarily from the mechanical interplay between the cells and between cells and the substrate.

### Surface adhesion drives ordering through differential growth.
In molecular liquid crystals, a lower temperature favors order due to the entropic driving force[29]. For out-of-equilibrium systems, such as growing biofilms, the driving force for ordering is more complex. We hypothesized that radial organization arises from the mechanical coupling between cells through their self-generated flow field, inspired by the alignment of rod-shaped objects under fluid shear[37,38]. Note that biofilm-dwelling cells are nonmotile; flow in this context is generated through cell growth and cell–cell interactions. To test our hypothesis, we tracked cell orientations and trajectories during biofilm development by using strains expressing a single intracellular punctum (Fig. 2a–e, Supplementary Fig. 8, and Supplementary Video 3)[16]. As WT* biofilms grew, cells towards the center tilted away from the substrate, developing a core of verticalized cells that expanded over time (Fig. 2c). The resulting growth-induced flow field had a zero-velocity core (Fig. 2a, d), corresponding to verticalized cells that project their offspring into the third dimension. In contrast,

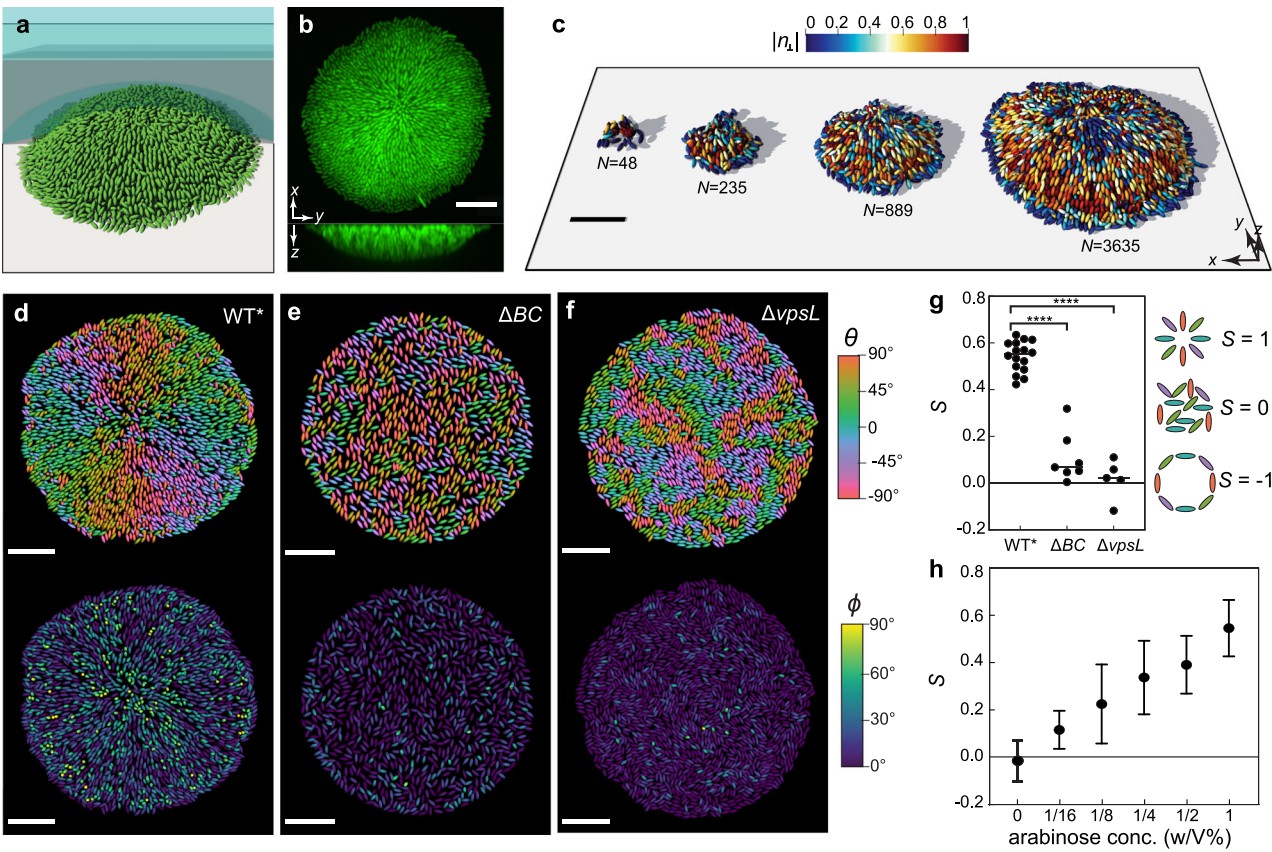

**Fig. 1 V. cholerae biofilms self-organize into aster patterns. a** Schematic of the experimental setup, where *V. cholerae* biofilms (green) were grown on a glass surface covered by a hydrogel (blue shaded). **b** Representative cross-sectional views of a WT* biofilm expressing mNeonGreen. **c** Single-cell 3D reconstruction of biofilm structures over time with different numbers of cells *N*. **d–f** Cell orientation color-coded according to each cell's angle in the basal plane $\theta$ (top) or the angle it makes with the substrate $\phi$ (bottom), in a biofilm that produces both exopolysaccharides and surface adhesion proteins (WT*; **d**), in a biofilm that only produces exopolysaccharides ($\Delta BC$; **e**), and in a bacterial colony with neither exopolysaccharides nor surface adhesion proteins ($\Delta vpsL$; **f**). Scale bars, 10 μm. **g** Radial order parameter *S* quantifying the degree to which cells conform to an aster pattern in the three strains (lines correspond to median values). Data were subjected to ANOVA with Bonferroni correction for comparison of means (**** denotes $P < 0.0001$; $P = 2 \times 10^{-11}$ for WT* vs. $\Delta BC$, $P = 6 \times 10^{-12}$ for WT* vs. $\Delta vpsL$ and $P = 0.19$ for $\Delta BC$ vs. $\Delta vpsL$). **h** *S* in biofilms in which the expression of *rbmC* is controlled by an arabinose inducible promotor (mean ± s.d.; $n \geq 4$ independent biofilms for each condition). Source data are provided as a Source Data file.

in the nonadherent mutant, the velocity field simply scaled linearly with the radial position. From the measured velocity field, we extracted the apparent in-plane proliferation rate *g* (Fig. 2b, d, inset and Supplementary Fig. 9). We found that *g* was uniform in the nonadherent biofilm: all cells in the basal layer were predominantly parallel to the substrate and therefore contributed to the basal layer expansion. In contrast, in the WT* biofilm, a growth void ($g \approx 0$) emerged in the center, with nearly uniform growth in the outer growing rim. Concomitant with the initiation of differential growth, cells aligned in an aster pattern, marked by a growing $S(r)$ with a rising peak (Fig. 2e).

**A reorientation cascade governs biofilm self-patterning.** We hypothesize that a mechanical synergy between cell verticalization, growth-induced flow, and aster pattern formation propels a reorientation cascade for biofilm self-patterning. On the one hand, cell-to-surface adhesion coupled with growth-induced mechanical stresses leads to stably anchored, verticalized cells in the biofilm center, which results in differentially oriented proliferation. On the other, differential proliferation drives cellular flows that radially align the cells in the rim. The radially expanding rim continuously accumulates stresses in the core of the biofilm, leading to cell verticalization and core expansion.

Below, we analyze the dynamic interplay of these two reorientation processes.

Step 1: To illustrate the formation and stabilization of the verticalized core, we consider a reduced problem consisting of a spherocylindrical cell that is parallel and adhered to a substrate and squeezed by its neighboring cells (Supplementary Information Section 3). The resulting energy landscape displays two distinct mechanical instabilities (Fig. 2f). The first instability corresponds to the verticalization event reported earlier[36,39–42]. Briefly, cells in a growing population mechanically push against one another, generating pressure. This pressure accumulates and eventually overcomes cell-to-substrate adhesion, causing cells to verticalize by rotating away from the substrate. The second instability corresponds to the pinch-off of these verticalized cells. In this case, neighboring cells generate forces in the out-of-plane direction, causing ejection of the verticalized cells from the substrate. For WT* cells, our analysis shows that the pinch-off of an already-verticalized cell is energetically more costly than the verticalization of a horizontal cell. Therefore, in a mixed population of verticalized and horizontal cells at a similar pressure, horizontal cells will preferentially verticalize before already-verticalized cells pinch-off. The reduction in basal footprint after verticalization reduces the total pressure inside the basal plane[36], causing verticalized cells to stably inhabit the

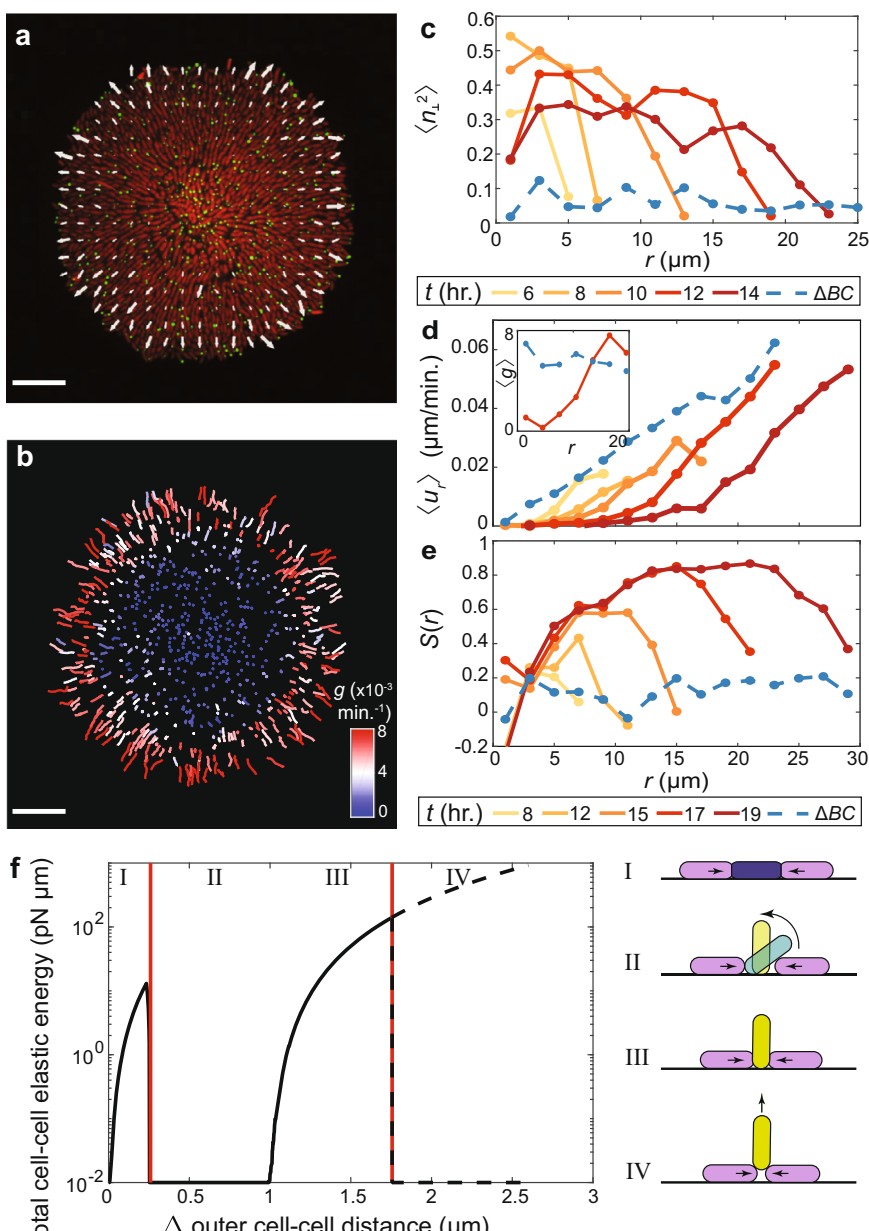

**Fig. 2 Growth-induced cellular flow and surface anchoring jointly lead to aster formation in biofilms.** **a** Raw image of the basal layer of a biofilm consisting of cells constitutively expressing mScarlet-I cytosolically and mNeonGreen-labeled puncta. Overlain is the velocity field measured from puncta trajectories. **b** Puncta trajectories colored by the apparent in-plane growth rate $g$. The apparent in-plane growth rate is calculated as $g(r) = (\partial_r r u_r)/r$ in a neighborhood around each cell. Scale bars, 10 μm. **c** Azimuthally averaged degree of verticalization $\langle n_\perp^2 \rangle$ as a function of distance $r$ from the center, in the basal layer. This quantification was performed in cells expressing mNeonGreen cytosolically, which allows for more accurate quantification of $n_\perp$. **d, e** Azimuthally averaged radial velocity $\langle u_r \rangle$ (Inset: apparent in-plane growth rate $\langle g \rangle \times 10^{-3}$ min.$^{-1}$) (**d**) and radial order parameter $S$ (**e**) as a function of distance $r$ from the center, in the basal layer. In **c–e**, the dashed blue lines denote results from the nonadherent mutant. **f** Results of a reduced problem showing the strain energy due to cell-to-cell contacts in a cell as it is squeezed by its neighbors (black line). The dashed red lines denote the results from stability analyses (Supplementary Information Section 3). Upon increasing compression, the central cell evolves through four phases, which are given schematically. Source data are provided as a Source Data file.

basal plane. This process preferentially occurs near the center of the biofilm where pressure is relatively high[36,40], leading to an accumulation of verticalized cells and therefore an expanding verticalized core. The smaller the cell-to-surface adhesion, the more favorable that pinch-off becomes (Supplementary Fig. 10), leading to less stable verticalized cells. These different dynamics can be seen by examining the differential puncta trajectories in the basal plane in the $\Delta BC$ and WT* mutants. In the $\Delta BC$ mutant, we notice the disappearance of puncta as cells are ejected from the basal layer, whereas in the WT* mutant, all puncta

remain in the basal layer (Supplementary Fig. 11). Similarly, distinct cell trajectories were also observed in the agent-based model (Supplementary Fig. 12). Since rod-shaped cells grow and divide along their long axes, this spatial segregation of cell orientation leads to spatially patterned differential growth.

Step 2: Next, we employ active nematic theory[21,38,43] to elucidate how differential growth can induce radial alignment. Defining the nematic order parameter $\mathbf{Q} = 2\langle \hat{\mathbf{n}}_\parallel \otimes \hat{\mathbf{n}}_\parallel - \mathbf{I}/2 \rangle$ as the head-tail symmetric tensor of cell orientation, mesoscopically

averaged over a small region ($\mathbf{I}$ is the identity matrix) its evolution in a surrounding flow $\mathbf{u}$ is governed by[44]

$$(\partial_t + \mathbf{u} \cdot \nabla)\mathbf{Q} - \Gamma\mathbf{H} = \lambda\mathbf{E} + \boldsymbol{\omega} \cdot \mathbf{Q} - \mathbf{Q} \cdot \boldsymbol{\omega}, \qquad (1)$$

where the right-hand side quantifies the driving force for the rod-shaped particles to rotate within a velocity gradient field. Here, $\mathbf{E} = \frac{1}{2}\left[\nabla\mathbf{u} + \nabla\mathbf{u}^T - (\nabla \cdot \mathbf{u})\mathbf{I}\right]$ is the traceless strain-rate tensor, $\boldsymbol{\omega} = \frac{1}{2}(\nabla\mathbf{u} - \nabla\mathbf{u}^T)$ is the vorticity tensor, which vanishes in the axisymmetric geometry being considered, and $\lambda$ is the flow-alignment parameter. For rod-shaped objects $\lambda > 0$, corresponding to a tendency for the rods to align with flow streamlines[17]. Finally, the nematic alignment term $\Gamma\mathbf{H}$ relaxes $\mathbf{Q}$ toward a bulk state with minimal angular variation, however, its contribution in biofilms is expected to be negligible since cells are buffered from each other by soft exopolysaccharides (see discussions in Supplementary Information Section 4). Indeed, in suspensions of rods, activity alone can generate nematic order through flow-alignment-coupling, in the absence of any passive elastic alignment[45,46].

Assuming axisymmetry, the evolution of the cell orientation field is given by[19,38]

$$\partial_t\Theta + u_r\partial_r\Theta = -f(r,t)\sin(2\Theta), \qquad (2)$$

where $\Theta$ is the angle between the local orientation field and the radial direction, $f = (\lambda r/4q)\partial_r(u_r/r)$ quantifies the aligning torque due to gradients in the flow field, and $q$ quantifies the degree of local ordering (Supplementary Information Section 4 and Supplementary Fig. 13). From $\partial_t\Theta \sim -f\sin(2\Theta)$, we find that a nonzero $f$ causes cells to rotate, and the direction of rotation is critically dependent on the sign of $f$.

Unlike passive liquid crystals, biofilm-dwelling cells generate their own velocity field through growth. Assuming uniform density, mass conservation requires $\nabla \cdot \mathbf{u} = g(r)$. In nonadherent mutant biofilms and bacterial colonies, growth is exclusively in-plane with a uniform growth rate $\gamma$, resulting in a linear velocity field, $u_r = \gamma r/2$, and thus a vanishing driving force for cell alignment ($f = 0$). Under this condition, cells are simply advected outwards without any tendency to align, leading to a macroscopically disordered pattern. In contrast, in WT* biofilms, verticalization stabilizes an expanding in-plane growth void of radius $r_0(t)$. This corresponds to a differential growth rate $g(r)$: 0 for $r \leq r_0$ and $\gamma$ for $r > r_0$. The resulting velocity field is $\gamma(r - r_0^2/r)/2$ for $r > r_0$, leading to a strictly positive driving force for radial alignment, $f = \frac{\lambda\gamma r_0^2}{4qr^2} > 0$, in the outer growing rim. In this case, $\Theta$ dynamically approaches 0, characteristic of an aster pattern (Supplementary Fig. 14). In fact, long-range order can be induced whenever a 2D growing bacterial collective deviates from an isotropically expanding pattern, for instance when confined in a rectangular geometry[47,48] or during inward growth[38]. This model thus reveals that differential growth, established by a verticalized core ($r_0 \neq 0$), generates the driving force for radial alignment in a growing biofilm. This driving force vanishes in the absence of a verticalized core ($r_0 = 0$), leading to a disordered phenotype.

**Imposing a growth void reproduces radial ordering.** A key prediction of the active nematic theory is that a growth void is sufficient to induce radial organization. To test this prediction, we patterned a growth void into an otherwise disordered biofilm. Specifically, we started with a nonadherent biofilm already grown for 17 hours and used a 405 nm laser to selectively kill the cells in the center. The vestiges of the dead cells sustained a growth void (Supplementary Fig. 15), mimicking the verticalized core in the WT* biofilm. Consistent with our model prediction, the

proliferating cells aligned radially over time in biofilms with a growth void, whereas biofilms without a growth void remained disordered (Fig. 3a–c). Conversely, our theory predicts that excess growth at the biofilm center should lead to $f < 0$ and therefore to vortex formation (Supplementary Information Section 4). Indeed, in another set of experiments, we observed that the nonadherent cells aligned circumferentially when excess growth was introduced at the center (Supplementary Fig. 16). We also quantitatively tested the validity of the model by prescribing a growth void with a fixed radius $r_0$ in a set of simplified 2D agent-based models (Supplementary Fig. 17 and Supplementary Video 4). We found that the instantaneous angular velocity of individual cells scaled linearly with $\sin(2\Theta)/r^2$ and increasing $r_0$ led to a quadratic increase in the angular velocity, all in agreement with the theory (Fig. 3d). Note that in both simulations and experiments, the radial order quickly saturated in the patterned biofilm with a fixed $r_0$, since the aligning force decays with $1/r^2$ as cells are advected outward. Thus, a growing $r_0(t)$ is necessary to reinforce radial alignment during biofilm expansion. This is indeed the case in WT* biofilms: growth of the outer rim accumulates pressure to generate more verticalized cells and expand the verticalized core, which in turn continuously drives alignment in the outer horizontal cells. To interrogate the mechanical interplay between these reorientation processes, we next develop a minimal physical model coupling verticalization of individual cells to the long-range radial ordering.

**Two-phase model of cell organization.** We decompose the biofilm into populations of two phases with vertical and horizontal cells and take the phase fractions to be $\rho$ and $1 - \rho$, respectively[36]. In this decomposition, we consider all cells that do not generate offspring in the basal plane as vertical, which does not strictly require $n_\perp = 1$. As a result, these vertical cells have some non-vanishing basal projection of their director field, $\mathbf{n}_\parallel \neq 0$, and therefore a well-defined $\mathbf{Q}$. The growth kinetics of the phases are governed by

$$\partial_t\rho + \nabla \cdot (\mathbf{u}\rho) = C(p)(1 - \rho), \qquad (3a)$$

$$\partial_t(1 - \rho) + \nabla \cdot (\mathbf{u}(1-\rho)) = \gamma(1 - \rho) - C(p)(1 - \rho). \qquad (3b)$$

Here, we assume that the horizontal-to-vertical conversion is driven by the local pressure $p$, where $C(p)$ is the conversion rate. We further assume that pressure arises from friction with the substrate $\nabla p = \eta\mathbf{u}$, where $\eta$ is the friction coefficient, and that only the horizontal cells generate growth in the basal layer, $\nabla \cdot \mathbf{u} = \gamma(1 - \rho(\mathbf{r}))$. Combined with Eq. (2), these equations generate a complete continuum description of the dynamics of cell growth and reorientation in the basal layer of biofilms (Supplementary Information Section 4). In this model, we have omitted any feedback of the orientation field onto the biofilm expansion. This is supported by the nearly isotropic local stress field predicted by our 2D agent-based model, which suggests that the radial alignment induced feedback is weak (Supplementary Fig. 18).

Numerical solutions of the model quantitatively reproduce the cascade of self-organization events (Fig. 4a–d), showing the intimate spatiotemporal coupling between cell verticalization and radial alignment. Many salient features of the experimental results are recapitulated by the model: for example, $S(r)$ reaches a maximum near the verticalized core where the driving force is the strongest. Interestingly, the model reveals a frozen splayed core where cells cease to reorganize (compare Figs. 2e and 4d): as the in-plane velocity goes to zero, the driving force to rotate also vanishes—cells in the core are thus locked as a fossil record that memorizes the mechanical history they have experienced.

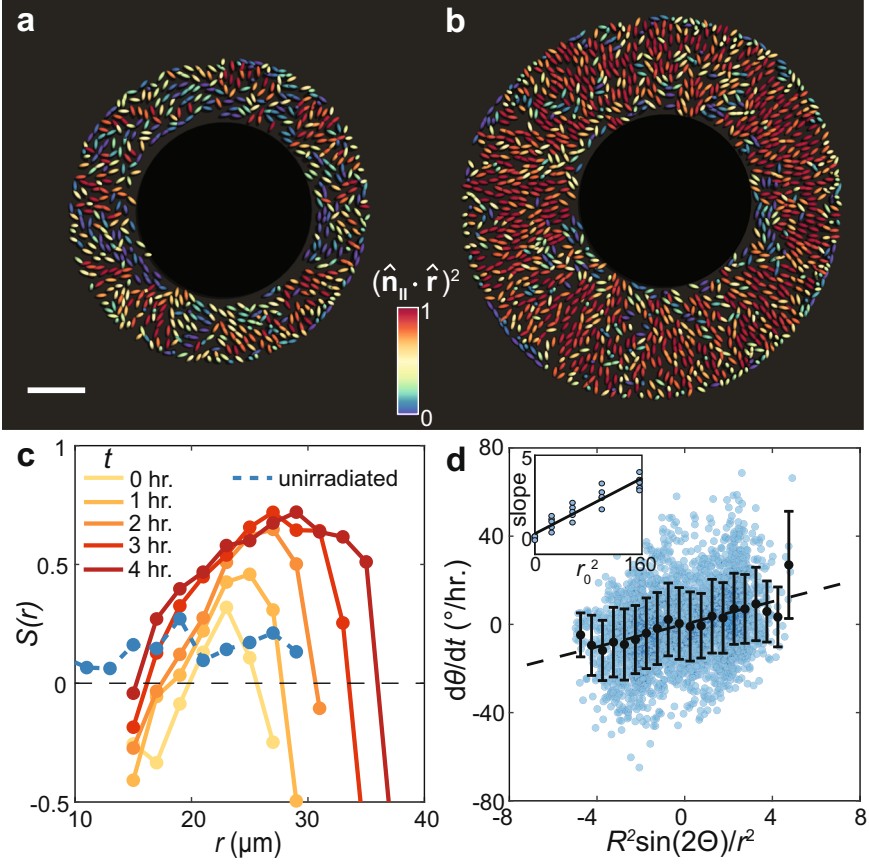

**Fig. 3 Cell organization can be manipulated by controlling spatial growth patterns. a, b** A nonadherent biofilm grown for 17 h was irradiated using 405 nm laser to induce cell death in a circle of radius 15 μm at the center. Colors denote the degree of radial alignment of individual cells $(\hat{n} \cdot \hat{r})^2$ 0 hours (**a**) and 4 hours (**b**) after irradiation. **c** $S(r)$ in the irradiated biofilm (colored according to time) and the unirradiated control (blue). **d** Angular velocity of individual cells from ABSs with a growth void plotted against the predicted nondimensionalized driving force (superimposed error-bars correspond to the mean ± s.d. of the data binned on 0.5-unit intervals; $n \geq 6$ cells per bin). Inset: fitted slope from (**d**) for different growth void sizes $r_0$ (μm²). Source data are provided as a Source Data file.

Importantly, the model yields robust results: regardless of the initial conditions and choice of parameters (Supplementary Figs. 19, 20), a WT* biofilm always patterns itself following the sequence shown in Fig. 4e. Our two-phase active nematic model thus elucidates the reproducible mechanical blueprint that guides the development of biofilm architecture.

## Discussion

To conclude, our results reveal a biomechanically driven self-patterning mechanism in bacterial biofilms in which cells synergistically order into an aster pattern. Specifically, we showed that stable cell verticalization at the core directs radial cell alignment in the rim during surface expansion. This inter-dependent reorientation cascade involves biofilm-wide, mechanical signal generation and transmission, in contrast to the bio-chemical signaling widely observed in other living organisms. In the first step of this cascade, growth-induced stresses and surface adhesion jointly cause cells in the core to verticalize but remain anchored to the substrate. Although cell verticalization has been studied in the literature using both microscopic[36,39–41] and mesoscopic[49–51] approaches, here we show that after verticali-zation, cells that secrete adhesins preferentially stay attached to the surface. These stably verticalized cells are the key to gen-erating differentially directed growth, which drives the radial alignment of the cells at the rim, the second step of the reor-ientation cascade. The aster-like pattern has also been observed in *V. cholerae* biofilms in other geometries and even in other

bacterial species[14,42], but how bacteria self-organize into such patterns has remained unexplained. Our single-cell imaging, agent-based simulations, and two-phase active nematic model together uncover the biomechanical mechanism responsible for self-organized pattern formation in bacterial biofilms. In *On Growth and Form*[52], D'Arcy Thompson wrote: "… growth [is] so complex a phenomena…rates vary, proportions change, and the whole configuration alters accordingly." Although over a century old, this statement still rings true today.

The long-range organization of biofilms may find several possible biological implications. Radial cell alignment could aid in nutrient diffusion toward the innermost regions of the biofilm, critical to biofilm survival and development. Indeed, molecular diffusion in liquid crystals has been shown to be anisotropic and faster along the nematic director[53]. Radial cell alignment could also facilitate cell dispersal[54] by lowering the escape barrier, that is, a smaller area of the extracellular matrix needs to be degraded for the cells to leave the biofilm. The radially aligned cells at the rim are also pre-positioned in an orientation optimal for swim-ming away. Finally, strong radial alignment can result in aniso-tropic stresses that are higher in the radial direction, which could potentially reinforce the self-patterning process and lead to more verticalized cells at the core to explore 3D space.

The biofilm self-patterning process relies critically on the sponta-neous flows generated through cell growth. Spontaneous flow gen-eration is a common phenomenon in various developmental systems, including ventral furrow formation in Drosophila[24], zebrafish

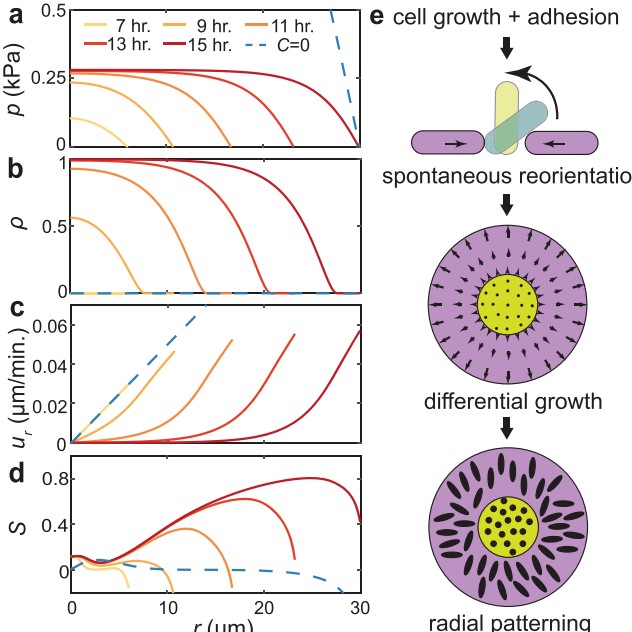

**Fig. 4 A two–phase active nematic model predicts the spontaneous generation of differential proliferation and macroscopic cell organization. a–d** Numerical solution of the model consisting of a population of horizontal and vertical cells. The biofilm was initiated with no vertical cells and random in-plane orientations. Evolution of pressure $p$ (**a**), fraction of vertical cells $\rho$ (**b**), in-plane radial velocity $u_r$ (**c**), and radial order parameter $S$ (**d**). Curves are colored according to time. Results for a biofilm that cannot sustain verticalized cells ($C = 0$) are shown in blue. **e** Schematic representation of the biofilm self-patterning process. Source data are provided as a Source Data file.

embryonic development[25], etc. While flow causes bulk morphological changes in these systems, in biofilms it acts to transmit mechanical forces and drive the long-range organization. It is intriguing to contemplate whether the synchronous mechanical coupling observed in *Vibrio cholerae* biofilms could be generalized to other organisms consisting of polarized cells that grow anisotropically. In a broader context, cell polarity and organization critically underlie collective cell function and normal development, as exemplified by topological defects that mediate 2D-to-3D transitions in motile *Myxococcus xanthus* colonies[55] and cell death and extrusion in epithelial layers[56]. Our findings hence shed light on the biomechanical control of cell organization through the spatiotemporal patterning of growth and pave the way to controlling cell organization by encoding synthetic biological circuits or through optogenetic manipulation[57].

## Methods

**Bacterial strains and cell culture.** All *V. cholerae* strains used in this study were derivatives of the streptomycin-resistant variant of the WT O1 El Tor biotype C6706str2[58] and contained a missense mutation in the *vpvC* gene ($vpvC^{W240R}$), which resulted in constitutive biofilm production through the upregulation of c-di-GMP (rugose/Rg strain)[27]. For the majority of the results presented in this work, we used a strain in which the gene encoding the cell-to-cell adhesion protein RbmA was deleted to minimize the effects of intercellular adhesion[15]; however, we found that our analysis equally applied to the rugose strain (Supplementary Fig. 1). We primarily worked with two other mutants: (1) ΔBC which included additional deletions of *bap1* and *rbmC* genes[31–33], and (2) Δ*vpsL* in which a key exopoly-saccharide biogenesis gene was deleted in the rugose background (RgΔ*vpsL*)[34,35]. In the absence of *bap1* and *rbmC*, the ΔBC mutant cells were unable to adhere to the substrate (referred to as the nonadherent mutant throughout the text). In the absence of *vpsL*, the cells did not properly synthesize exopolysaccharides and consequently, all accessary matrix proteins, which bind to the exopolysaccharide, did not function properly[28]. For velocity field measurements, we used strains containing the μNS protein from the avian reovirus fused to an mNeonGreen fluorescent protein, which were shown to self-assemble into a single intracellular

punctum[16,59]. All strains used in the study were also modified to constitutively produce either mNeonGreen or mScarlet-I fluorescent proteins. Mutations were introduced using either the pKAS32 exchange vector[60] or the MuGENT method[61]. For a full list of strains used, see Supplementary Table S1. Biofilm growth experiments were performed using M9 minimal media supplemented with 0.5% glucose (w/w), 2 mM MgSO$_4$, 100 μM CaCl$_2$, and the relevant antibiotics as required (henceforth referred to as M9 media).

Experiments began by first growing *V. cholerae* cells in liquid LB overnight under shaken conditions at 37 °C. The overnight culture was diluted 30× in M9 media and grown under shaken conditions at 30 °C for 2–2.5 h until it reached an OD$_{600}$ value of 0.1–0.2. The regrown culture was subsequently diluted to an OD$_{600}$ of 0.001 and a 1 μL droplet of the diluted culture was deposited in the center of a glass-bottomed well in a 96-well plate (MatTek). Concurrently, agarose was dissolved in M9 media at a concentration of 1.5–2% (w/V) by microwaving until boiling and then placed in a 50 °C water bath to cool without gelation. After cooling, 200 nm far-red fluorescent particles (Invitrogen F8807) were mixed into the molten agarose at a concentration of 1% (V/V) to aid in image registration. Next, 20 μL of the molten agarose was added on top of the droplet of culture and left to cool quickly at room temperature, to gel, and to trap the bacterial cells at the gel-glass interface. Subsequently, 100 μL of M9 media was added on top of the agarose gel, serving as a nutrient reservoir for the growing biofilms. The biofilms were then grown at 30 °C and imaged at the designated times.

**Image acquisition.** Images were acquired using a confocal spinning disk unit (Yokogawa CSU-W1), mounted on a Nikon Eclipse Ti2 microscope body, and captured by a Photometrics Prime BSI CMOS camera. A ×100 silicone oil immersion objective (N.A. = 1.35) along with 488 nm, 561 nm and 640 nm lasers were used for imaging. This combination of hardware resulted in an *x*–*y* pixel size of 65 nm and a *z*-step of 130 nm was used. For end-point imaging, biofilms were imaged after 12–24 h of growth, and only the 488-nm channel, corresponding to the mNeonGreen expressing cells, was imaged. For time-lapse imaging, samples were incubated on the microscope stage in a Tokai Hit stage top incubator while the Nikon perfect focus system was used to maintain focus. Images were captured every 30 min, and in addition to the 488-nm channel, the 640-nm channel was used to image the fluorescent nanoparticles.

For velocity measurements, cells constitutively expressing mScarlet-I cytosolically and mNeonGreen-labeled puncta were imaged using a slightly modified procedure. The 488-nm channel, corresponding to the puncta, was imaged every 2–10 min while the 561-nm channel, corresponding to the cells, was imaged every 1–2 h. This procedure allowed us to image the relatively bright puncta with low laser intensity and therefore minimal photobleaching and phototoxicity, as high temporal resolution is required to accurately track puncta motion. To further reduce photobleaching and phototoxicity, we used a *z*-step of 390 nm when imaging the puncta. When imaging the cells, a *z*-step of 130 nm was used in the mScarlet-I channel to sufficiently resolve the position and orientation of the cells. Note that since mScarlet-I fluoresces at a longer wavelength than mNeonGreen and the corresponding signal-to-noise ratio was lower, the resulting images had poorer *z*-resolution and contrast and therefore resulted in greater uncertainties in $n_\perp$ measurements. We also restricted our attention to the basal flow field and therefore only imaged the bottom 3 μm of each biofilm. All images shown are raw images rendered by Nikon Elements software unless indicated otherwise.

**Overview of image analysis.** Raw images were first deconvolved using Huygens software (SVI) using a measured point spread function. The deconvolved three-dimensional confocal images were then binarized, layer by layer, with a locally adaptive Otsu method. To accurately segment individual bacterium in the densely packed biofilm, we developed an adaptive thresholding algorithm. For more details see Supplementary Information Section 1. Once segmented, we extracted the cell positions by finding the center of mass of each object, and the cell orientations by performing a principal component analysis. The positions and directions of each cell were converted from cartesian ($x, y, z, \hat{n}_x, \hat{n}_y, \hat{n}_z$) to cylindrical polar ($r, \psi, z, \hat{n}_r, \hat{n}_\psi, \hat{n}_z$) coordinates where the origin was found by taking the center of mass of all of the segmented cells in the ($x, y$) plane. We define the out-of-plane component of the direction vector as $n_\perp = \hat{\mathbf{n}} \cdot \hat{\mathbf{z}}$ and the in-plane component as $\mathbf{n}_\parallel = \hat{\mathbf{n}} - (\hat{\mathbf{n}} \cdot \hat{\mathbf{z}})\hat{\mathbf{z}}$, which we normalize as $\hat{\mathbf{n}}_\parallel = \mathbf{n}_\parallel/|\mathbf{n}_\parallel|$. Reconstructed biofilm images were rendered using Paraview.

**Measurement of the growth-induced velocity field.** To measure the growth-induced velocity field we used particle-tracking velocimetry on the puncta trajectories. The deconvolved puncta images were first registered using Matlab built-in functions. Puncta were detected by first identifying local intensity maxima in the 3D images, and sub-pixel positional information was found by fitting a para-bola to the pixel intensity around the maxima. This procedure was repeated for all frames yielding puncta locations over time which were then connected from frame to frame using a particle-tracking algorithm[62]. The radial velocity $u_r$ was calculated by fitting a straight line through the time vs. radial displacement data over a time interval of 1 h.

**Opto-manipulation of cell growth**. Previous work has shown the bactericidal effects of high energy, near-UV light[63]; therefore, we used spatially patterned 405 nm light to kill a subset of cells within a biofilm. Specifically, an Opti-Microscan XY galvo-scanning stimulation device with a 405 nm laser was used to selectively illuminate and kill cells within a cylindrical region at the center of the biofilm. We verified cell killing by staining the sample with propidium iodide (Supplementary Fig. 15). The same procedure used to measure the growth-induced velocity field (see above) was applied to the irradiated and control biofilms to measure cell orientation and trajectory dynamics simultaneously.

**3D agent-based simulations**. Building on the agent-based simulations developed by Beroz et al.[36] and others[40,64,65], we modeled cells as spherocylinders with a cylinder of length $L(t)$ and two hemispherical caps of radius $R$. The growth of each cell was assumed to be unidirectional and exponential, where the growth rate $\gamma$ was normally distributed with a mean of $\gamma_0$ and a standard deviation of $0.2\gamma_0$. Here, noise was added to account for the inherent stochasticity in cell growth and division. Each cell elongated exponentially until its length reached $L_{\max} = 2L_0 + 2R$, at which point it was replaced by two daughter cells with the length $L_0$. The doubling time can be calculated to be $t_{\text{double}} = \frac{1}{\gamma}\log\left(\frac{10R+6L_0}{4R+3L_0}\right)$. The cell-to-cell and cell-to-substrate contact mechanics were described by linear elastic Hertzian contact mechanics[66], with a single contact stiffness $E_0$; note that $E_0$ corresponds to the modulus of the soft exopolysaccharide in the matrix (~$10^2$ Pa) rather than the cell itself, which is much stiffer (~$10^5$ Pa). Correspondingly, the $R$ value we used (0.8 μm) is larger than the physical size of a cell (~0.4 μm). The parameter values we used were calibrated by rheological measurement and microscopy analysis, and have been shown to successfully capture the dynamics of biofilm-dwelling cells in prior work[36]. The cell-to-substrate adhesion energy was assumed to be linear with the contact area, with adhesion energy density $\Gamma_0$. We incorporated two viscous forces to represent the motion of biofilm-dwelling cells at low Reynold's number: (1) a bulk viscous drag for all degrees of freedom, and (2) a much larger in-plane surface drag for cells near the substrate, representing the resistance to sliding when a cell is adhered to the substrate via the surface adhesion proteins RbmC/Bap1. The two damping forces ensured that the cell dynamics were always in the overdamped regime.

We treated the confining hydrogel as a homogenous, isotropic, and linear elastic material using a coarse-grained approach. The geometry of the coarse-grained gel particles was assumed to be spherical with a radius $R_{\text{gel}}$. The interaction between particles was modeled using a harmonic pairwise potential and a three-body potential related to bond angles. The contact repulsions between the gel particles and the cells as well as between the gel particles and the substrate were described using linear elastic Hertzian contact mechanics. We treated the adhesion between the gel and the substrate using a generalized JKR contact model[67] and we also included a small viscous damping force to ensure the dynamics remained overdamped. The hydrogel was initialized by annealing the system to achieve an amorphous configuration.

Simulations were initialized with a single cell lying parallel to the substrate and surrounded by gel particles. Initially, a small hemispherical space surrounding the cell was vacated to avoid overlap between the cell and the hydrogel particles. We fixed a small number of hydrogel particles near the boundaries to provide anchoring for the elastic deformation of the hydrogel; the boundaries were kept sufficiently far away from the biofilm to minimize any boundary effects. We applied Verlet integration and Richardson integration to numerically integrate the equations of motion for the translational and rotational degrees of freedom, respectively. We implemented the model based on the framework of LAMMPS[68], utilizing its built-in parallel computing capability. For a more detailed description on the ABS, see Supplementary Information Section 2.

**Quasi-2D agent-based simulations**. To further verify the alignment dynamics of the continuum model quantitatively (Eq. (2), main text), we developed a set of quasi-2D simulations to mimic the laser irradiation experiments. To simplify the system, the translational and rotational degrees of freedom related to the vertical direction were set to zero, while all other parameters were kept the same as the 3D simulations. In each simulation, the bacteria first proliferate normally for 12 h, at which point the growth rate of the cells within a radius $r_0$ from the center of the biofilm was set to 0, mimicking the zone of dead cells caused by laser irradiation (Supplementary Fig. 17). In agreement with experiments, the simulated biofilm was initially randomly oriented ($S \approx 0$); however, cells tended toward an aster pattern and $S$ increased over time when the growth void was introduced. The predicted rate at which the cells were driven toward this pattern, in the Lagrangian frame of reference of the cells, is $D_t\Theta = -\frac{\lambda\gamma r_0^2}{4qr^2}\sin(2\Theta)$. We tested this relationship in the simulation data by comparing the angular velocity $D_t\Theta$ and $\frac{1}{r^2}\sin(2\Theta)$ (Fig. 3d). Note that we nondimensionalized the x-axis by the final colony radius 25 μm. We varied the radius of the growth void $r_0$ and repeated the same procedure and for each simulation run, we plotted the slope of the line of best fit versus $r_0^2$ (Fig. 3d, inset).

**Statistics**. As per convention in the field, standard $t$ tests were used to compare groups; the details of each test are described in the figure captions. No statistical method was used to predetermine sample size. No data were excluded from the analyses. The experiments were not randomized. The Investigators were not blinded to allocation during experiments and outcome assessment.

## Data availability

All relevant data supporting the key findings of this study are available within the article and its Supplementary Information files or from the corresponding author upon reasonable request. Source data are provided with this paper.

## Code availability

Matlab codes for single-cell segmentation[69] and the molecular dynamics code used for the agent-based simulations[70] are available online on Zenodo.

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

## Acknowledgements

We thank Drs. A. Mashruwala and Y. Xu for their help in the initial experiments. We thank Drs. S. Mao, T. Cohen, and J.-S. Tai for helpful discussions and B. Reed and M. Zhao for help with developing the ABSs. Research reported in this publication was supported by the National Institute of General Medical Sciences of the National Institutes of Health under Award Number DP2GM146253 (awarded to J.Y.). J.Y. holds a Career Award at the Scientific Interface from the Burroughs Wellcome Fund (1015763.02).

## Author contributions

J.N. and J.Y. conceived the project. J.N. and J.Y. designed and performed the experiments. J.N., Q.Z., H.L., C.L., S.Z. and J.Y. analyzed the data. C.L. and S.Z. developed the agent-based simulations. J.N. developed the continuum theory. J.N., C.L., S.Z., and J.Y. wrote the paper.

## Competing interests

The authors declare no competing interests.

## Additional information

**Peer review information** *Nature Communications* thanks Mehrana Raeisin Nejad and the other anonymous reviewer(s) for their contribution to the peer review this work. Peer reviewer reports are available.

