## [Peer Review File · Nature Communications]

Reviewers' Comments:

Reviewer #1:

Remarks to the Author:

The manuscript by Nijjer et al. reports a self-patterning program of growing *Vibrio cholerae* biofilms confined between glass and an agarose gel. Specifically, the authors found that *V. cholerae* cells that possess adhesion to the substrate surface self-organize into an aster pattern, which consists of a core of verticalized cells and an outer rim of radially aligned cells. In contrast, this radial order is destroyed in nonadherent cells. The authors further hypothesized that this global ordering results from two synergetic steps: (1) growth-induced mechanical pressure causes cells in the biofilm to verticalize, and (2) verticalized cells lead to spatially nonuniform growth and expansion that mechanically align outer cells in the radial direction. The authors combined various experimental and theoretical tools to verify their hypothesis. In particular, the authors combined agent-based simulations and continuum theory to show that a minimal mechanical model is sufficient to recapitulate the observed patterning/ordering dynamics. In addition, the authors also used laser irradiation to introduce a growth void in the core of an otherwise disordered biofilm. This experiment shows that a manually imposed differential growth pattern is sufficient to drive the radial ordering.

This work will be a great addition to the existing literature on active matter and mechanomorphogenesis. In particular, this work uses a bacterial system to demonstrate how the interplay between mechanics and growth robustly drives the development of an ordered pattern. This work in *V. cholerae* also nicely contrasts many previous works studying *E. coli* grown under a similar setup. Unlike WT *V. cholerae* biofilms, *E. coli* biofilms tend to develop into layers of horizontally oriented cells (cf. Duvernoy et al. Nature Comms. 2018, Fig. 1). Moreover, each layer develops a mosaic of locally ordered blocks (similar to the $\Delta vpsI$ mutant), presumably because their adhesion to the substrate surface is different. I think the comparison of the two systems is a great example of how differences in mechanical interaction with the environment can lead to drastically different global patterns and morphologies in bacterial colonies.

Major questions/comments/suggestions:

1. Does anisotropic growth affect the theoretical results? If rod-shaped cells are aligned radially, their growth should in principle be preferentially along the radial direction. This anisotropy is missing from the equation: $\nabla \cdot \mathbf{u} = \mathbf{g}$.
2. In the two-phase active nematic model, it seems that the nematic order does not feed back onto the velocity or pressure. Thus, the dynamics of colony growth and cell verticalization is independent of the radial ordering of cells. This contradicts some remarks by the authors (e.g. at lines 130-31 "On the other hand, differential proliferation drives cellular flows that radially align the cells in the rim, which in turn facilitates cell verticalization and core expansion." This point requires clarification both in the main text and SI.
3. It's not immediately obvious why vertical cells are more deeply embedded than horizontal cells. First, it's not clear what energy function the authors are using for this calculation, and this should be clarified. Second, it would be very helpful if the authors could provide some intuition for this observation about relative embedding depth. On a related note, I am not sure whether plotting the cell-cell elastic energy as done in fig. 2f is the correct way to demonstrate the relative order of the two instabilities. Denote by r the contact radius between the "instability" cell and the substrate, and by L the cell length, the base-layer contour length for a horizontal cell configuration is $2rL$ while it takes a different value, πr^2 , for a vertical cell configuration. Thus, considering a constant number of two cells around the verticalized cell seems biased. I think a better way is to compare the critical pressure p_c for these two instabilities to happen -- this will correct for the contour length difference, since p_c times the contour length will give the total force exerted on the "instability" cell. The authors should implement this alternative approach and see if it still predicts the observed order of instabilities, or if the order is simply unavoidable - i.e. cells must first verticalize before they can detach from the substrate.
4. Most experiments were done in a setup where the biofilms are confined between glass and

agarose gels. What's the role of this confinement? It's unclear whether the confinement is essential for the development of the reported spatial ordering. If not, does the proposed mechanism also apply to a submerged biofilm grown on glass (at least to the basal layer of cells)?

5. Are there any biological consequences of developing the radially ordered pattern? For example, would it help the cells to escape confinement when the biofilm disassembles? The manuscript would benefit from additional discussion along these lines.

Minor questions/comments/suggestions:

6. For the general reader, the authors should show a schematic of cell being reoriented by a local 1D velocity gradient. Along these lines, it would also be helpful to clarify that in Eq. 1 for the geometry considered $\omega=0$.

7. Line 85-88: Will overexpression of RbmC and/or Bap1 to an even higher level diminish the reported pattern? Presumably with extremely strong surface adhesion, the cells will fail to verticalize and trigger the cascade of reorientation, right?

8. Line 112-114: It's unclear which feature of Fig. 2c shows "a verticalized core that expands over time". Specifically, the 15-19 h curves overlap with each other.

9. Line 230-232: In experiments, the cells in the basal layer could also experience forces exerted by cells in the layer above. Could the authors comment on how the 3D structure might affect the mechanism of radial ordering, and more broadly the entire patterning dynamics?

10. Fig.1a: The cells in the top surface of the biofilm also seem to possess a radial ordering. What is the origin of this ordering? Is it inherited from the order in the basal layer or does it arise via a distinct mechanism?

11. Fig.1f: The $\Delta vpsI$ mutant doesn't form the same aster pattern as the WT* presumably because the adhesion proteins Bap1 and RbmC do not accumulate around the mutant cells (Berk et al. Science 2012). Is there an additional role that VPS plays in the reported self-patterning program? I.e., if $\Delta vpsI$ mutant cells were grown on surfaces coated with Bap1 and RbmC proteins, would they develop the same aster pattern as WT*?

Reviewer #2

Remarks to the Author:

This manuscript reports investigations on spatial structures of growing *Vibrio cholerae* biofilms. The authors used a combination of experimental, numerical and theoretical tools. They demonstrated a collective cell reorientation cascade in growing *Vibrio cholerae* biofilms that leads to a differentially ordered, spatiotemporally coupled core-rim structure. Cell verticalization in the core generates differential growth that drives radial alignment of the cells in the rim, while the radially aligned rim in turn generates compressive stresses that expand the verticalized core. Agent-based simulations and two-phase active nematic modeling reproduce the experimental observations and reveal the driving mechanisms underlying the differential ordering.

In my opinion, the results in this well-written manuscript are novel and tell an interesting story. I recommend the publication of this manuscript after the authors address the following questions:

1. All results in the manuscript are limited to colony size about 30 μm in radius. What limits the colony size? Would the radially aligned rim persist in colonies of much larger size.
2. Some data show imperfect radial alignment. For example, one can see some chirality in Extended Data Fig.2 d. Where does this weak chirality come from?
3. What is the thickness of the agarose gel? Does the observed core-rim structure depend on the specific parameters of the agarose gel, such as thickness and elastic moduli?

The authors present experimental results on growing colonies of *Vibrio cholerae* biofilms and show that, in the presence of adhesion between cells and the substrate, the basal layer of the colony forms a core of vertical cells at the centre and cells at the periphery orient radially and stay in the layer. The authors then explain this behaviour by considering a radial flow created by the verticalized core. Since the bacteria are elongated, they align to the direction of shear and as such bacteria at the rim align radially due to the flow created by the core. The radial alignment of the cells in the periphery generates compressive stresses that expand the verticalized core and, as a result, there is a feedback loop between the core and the rim which stabilises the pattern.

To understand the behaviour observed in the experiment, the authors also show that the pattern can be captured by agent based simulations of elongated rods which grow and divide and have surface adhesion with the substrate. In the absence of surface adhesion, neither orientation order nor a vertical core is observed and flows are random. This is correct for both experiments and agent based simulations.

Finally, a continuum model with two populations of in-plane and vertical cells is introduced to clarify the reason behind the formation of the pattern. In the continuum model, the in-plane and vertical populations transfer to one another due to a pressure. For the population with in-plane director, a 2D (in-plane) orientation field is considered. It is then shown that flows produced by inhomogeneous growth can lead to either aster or vortex orientations of the in-plane cells (depending on the core being a void growth region or a region with an excess growth).

The results and experimental observations are interesting, but there are a few points that need further clarification.

1. Since the adhesion with a substrate plays an essential role in the formation of the pattern, it should be clarified how and why the presence and absence of surface adhesion leads to different behaviours. In the SM, it has been argued that there is a “pinch-off” instability upon which the cells get ejected from the layer. It is discussed that, in the presence of the surface adhesion, the energy cost of this instability (for the values of the parameters in this experiment), is higher than the verticalization instability, and this energy difference vanishes in the absence of surface adhesion. This leads to spontaneous ejection of non-adhesive cells upon verticalization and as such, there is no vertical core in the non-adhesive mutant.

In the paper in its current form, there is not enough evidence supporting the “pinch-off” instability. Considering the agent based simulations and the experimental data, there is no measurement in the ejected cells. Is there any evidence in the experiments/simulations which support the

ejection instability? In the experiment, if cells get ejected from the basal layer, the height of the colony should grow. As a result, comparing the height of the colony in the adhesive and non-adhesive case should show if such instability happens in the non-adhesive case or not.

If it is not possible to judge from the experimental data about the height (due to any reason), it should be possible to see such an instability in the simulation of the rods (for example formation of local second layer in the simulations). Could authors provide such data to support the instability?

2. There has been a considerable amount of interest in orientation of bacteria in biofilms recently. Verticalization of the bacteria has been also observed in other experiments (for example Ref. 32). The current manuscript lacks an overview and comparison between the experiments in this paper and other works that report formation of verticalized cores. As an other example formation of a vertical core with radially aligned rim is also observed in "Spatiotemporal establishment of dense bacterial colonies growing on hard agar", however the focus of that paper is mainly on the role of nutrient depletion in a biofilm.
3. The authors consider the vertical core as an in-plane growth void. Does it mean that the vertical cells do not divide? Or do they divide but the mass increase goes to the third direction? Is it possible to do measurements on the growth to the third direction?
4. The authors explain that the formation of the radial order in the rim is due to active flows and not due to passive interactions. It could be helpful to mention that the formation of nematic order due to active flows in a flow aligning regime has been studied before (See for example Sreejith Santhosh, et al, Journal of Statistical Physics,180 (2020)).
5. In a similar basis, recent studies have shown that the growth into the third direction and verticalization can be caused by active "extensile" flows (such as flows created due to cell division). See for example "Extensile stress promotes out-of-plane flows in active layers, Mehrana Nejad, et al." and "Defect-mediated morphogenesis, Ludwig A. Hoffmann, et al". These may explain the formation of the vertical core in a dividing colony. Although the rim seems to be in the isotropic phase, the collision and the interaction between bacteria seem to be essential for the formation of the vertical core.
6. Comparing Figs. 4b and 4d shows that the radial alignment parameter S is non-zero in the core, meaning that the core is not perfectly vertical but has a 3d splay form. This should be clarified and discussed in the main text.

minor comments:

1. The definition of \hat{n}_{\parallel} in the radial order parameter S is not clear. Going through the SI it becomes clear, but it would be helpful to have it in the main text.
2. The data in Fig. 3d) is widely spanned along y -axis. It could be helpful to show the average values (with error-bars) on top.
3. Figs. 4 a)-d) show the pressure, fractional density of vertical cells, radial velocity, and radial order of cells as a function of position in the colony in different times. It could be helpful to scale the x -axis with the size of the colony at different times to make the comparison easier.
4. In the line below Eq. 23 in the SM, λ needs to be larger than “zero” for rod-shape particles.

Reviewer #4:
None

Reviewer 1:

The manuscript by Nijjer et al. reports a self-patterning program of growing *Vibrio cholerae* biofilms confined between glass and an agarose gel. Specifically, the authors found that *V. cholerae* cells that possess adhesion to the substrate surface self-organize into an aster pattern, which consists of a core of verticalized cells and an outer rim of radially aligned cells. In contrast, this radial order is destroyed in nonadherent cells. The authors further hypothesized that this global ordering results from two synergetic steps: (1) growth-induced mechanical pressure causes cells in the biofilm to verticalize, and (2) verticalized cells lead to spatially nonuniform growth and expansion that mechanically align outer cells in the radial direction. The authors combined various experimental and theoretical tools to verify their hypothesis. In particular, the authors combined agent-based simulations and continuum theory to show that a minimal mechanical model is sufficient to recapitulates the observed patterning/ordering dynamics. In addition, the authors also used laser irradiation to introduce a growth void in the core of an otherwise disordered biofilm. This experiment shows that a manually imposed differential growth pattern is sufficient to drive the radial ordering.

This work will a great addition to the existing literature on active matter and mechanomorphogenesis. In particular, this work uses a bacterial system to demonstrate how the interplay between mechanics and growth robustly drives the development of an ordered pattern. This work in *V. cholerae* also nicely contrasts many previous works studying *E. coli* grown under a similar setup. Unlike WT* *V. cholerae* biofilms, *E. coli* biofilms tend to develop into layers of horizontally oriented cells (cf. Duvernoy et al. Nature Comms. 2018, Fig. 1). Moreover, each layer develops a mosaic of locally ordered blocks (similar to the $\Delta vpsL$ mutant), presumably because their adhesion to the substrate surface is different. I think the comparison of the two systems is a great example of how differences in mechanical interaction with the environment can lead to drastically different global patterns and morphologies in bacterial colonies.

Response: We thank the reviewer for their positive and constructive comments. We below address these comments in a point-by-point manner.

Major questions/comments/suggestions:

1. Does anisotropic growth affect the theoretical results? If rod-shaped cells are aligned radially, their growth should in principle be preferentially along the radial direction. This anisotropy is missing from the equation: $\nabla \cdot \mathbf{u} = g$.

Response: We thank the reviewer for bringing up this interesting argument. As the reviewer points out, in our theoretical model we assume that the growth in the basal plane is isotropic. This assumption is based on the fact that even though cells are radially aligned, as the biofilm expands, the expansion rate (i.e., growth) is still isotropic: the growing cells fill out 2D space without an obvious bias for the circumferential or radial direction, as reflected by the equation $\nabla \cdot \mathbf{u} = g$.

Although growth is isotropic, cell alignment could, in principle, result in locally anisotropic stresses (e.g. Doostmohammadi *et al.* PRL **117**, 048102. 2016). We do not include this anisotropy in our current work because the anisotropy is small and unlikely to change our overall findings.

Indeed, in the updated Extended Data Fig. 18 of the Supplementary Information, we have shown that the difference between the radial and azimuthal stress is less than 10% of the average stress. We agree with the reviewer that including this anisotropy might lead to a higher-order mechanical feedback that further reinforces cell ordering; however, this added complexity renders the theoretical model analytically intractable. We have incorporated this discussion in the updated manuscript.

2. In the two-phase active nematic model, it seems that the nematic order does not feed back onto the velocity or pressure. Thus, the dynamics of colony growth and cell verticalization is independent of the radial ordering of cells. This contradicts some remarks by the authors (e.g. at lines 130-31 “On the other hand, differential proliferation drives cellular flows that radially align the cells in the rim, which in turn facilitates cell verticalization and core expansion.” This point requires clarification both in the main text and SI.

Response: We thank the reviewer for pointing out our inaccurate wording in the original manuscript. Indeed, in our analytical model, the radial alignment does not feed back onto the pressure, the degree of verticalization, or velocity. Our original intent in the manuscript was to highlight the fact that the growing rim, regardless of whether it is ordered or not, does feed back onto cell verticalization by generating pressure. We agree with the reviewer that our original wording was confusing. We have now updated this sentence and clarify that we neglect this feedback in the two-phase model.

3. It's not immediately obvious why vertical cells are more deeply embedded than horizontal cells. First, it's not clear what energy function the authors are using for this calculation, and this should be clarified. Second, it would be very helpful if the authors could provide some intuition for this observation about relative embedding depth. On a related note, I am not sure whether plotting the cell-cell elastic energy as done in fig. 2f is the correct way to demonstrate the relative order of the two instabilities. Denote by r the contact radius between the "instability" cell and the substrate, and by L the cell length, the base-layer contour length for a horizontal cell configuration is $2rL$ while it takes a different value, πr^2 , for a vertical cell configuration. Thus, considering a constant number of two cells around the verticalized cell seems biased. I think a better way is to compare the critical pressure p_c for these two instabilities to happen -- this will correct for the contour length difference, since p_c times the contour length will give the total force exert on the "instability" cell. The authors should implement this alternative approach and see if it still predicts the observed order of instabilities, or if the order is simply unavoidable - i.e. cells must first verticalize before they can detach from the substrate.

Response: The energy functions used in the single-cell analysis are equivalent to the force functions used in the agent-based model. The key difference is that we restrict our attention to a minimal model considering the instability of a single cell. We now explicitly state the energy functions in the SI. The reason why the vertical cells have a larger penetration depth has to do with the difference in the geometry between the hemispherical cap and the cylindrical body, and how the cell-surface adhesion and cell-surface repulsion energies scale with penetration depth. We now give more details on how the results in SI section III are derived.

Next, we agree with the reviewer’s remark that by considering only two cells, we are not accounting for the contour length difference between the two configurations. Following the reviewer’s suggestion, we now plot the total work done on the “instability” cell assuming a hypothetical configuration in which this cell is completely surrounded by neighboring cells. In the updated model, we still find that for the WT* biofilm the total work required to verticalize a horizontal cell is smaller than to pinch-off an already-verticalized cell. This is because the difference in contour lengths between the two configurations is a factor of $\frac{2L+2\pi R}{2\pi R} = 1.4 - 2.4$ (depending on the instantaneous length of the cell) and is insufficient to account for the difference between the two instability energies.

Finally, we note that the selection between cell verticalization and cell pinch-off does not take place at the individual cell level, but at the population level. As correctly pointed out by the reviewer, a single cell must first verticalize before it can detach from the substrate. However, in a population of cells that are experiencing a similar pressure, our calculations show that horizontal cells will verticalize first before already-verticalized cells pinch off. This order leads to the observation of stably verticalized cells. We have further clarified this point in the updated text.

4. Most experiments were done in a setup where the biofilms are confined between glass and agarose gels. What’s the role of this confinement? It’s unclear whether the confinement is essential for the development of the reported spatial ordering. If not, does the proposed mechanism also apply to a submerged biofilm grown on glass (at least to the basal layer of cells)?

Response: The core-rim structure and overall radial alignment are also commonly observed in biofilms growing in the absence of confinement; see the figure below (Extended Fig. 4) as well as references (Hartmann *et al.*, *Nat. Phys.* **15**, 251-256. 2020; Warren *et al.* *eLife* **8**, e41093. 2019; Yan *et al.* *PNAS* **113**, e53375343. 2016). This is an observation that has remained unexplained in the literature, and we believe that our theory and simulations provide a clear explanation. The main role that confinement plays is to impart normal stresses that tend to flatten the biofilm, leading to an overall larger basal footprint and more pronounced ordering since a larger fraction of cells are at the biofilm-glass interface. It does not, however, change the basic biophysical mechanism of radial ordering revealed in the current manuscript. We have now clarified this point in the updated text.

Role of confinement. *a*, Cross-sectional view of the basal plane of a WT* biofilm grown without an overlain gel. *b*, *c*, The same biofilm reconstructed where cells are color-coded by the angle ϕ each cell makes with the substrate (*b*) and the degree of radial alignment $(\hat{n}_r \cdot \hat{r})^2$ (*c*). We found that cells in unconfined biofilms also tended to align radially ($S = 0.18 \pm 0.05, n = 4$), albeit to a lesser extent than confined biofilms. Note that cells that do not belong to this cluster are manually removed. Scale bars, 10 μm .

5. Are there any biological consequences of developing the radially ordered pattern? For example, would it help the cells to escape confinement when the biofilm disassembles? The manuscript would benefit from additional discussion along these lines.

Response: We thank the reviewer for this helpful suggestion. We have now dedicated a paragraph in the discussion section to the possible biological implications of the observed ordering pattern.

Minor questions/comments/suggestions:

6. For the general reader, the authors should show a schematic of cell being reoriented by a local 1D velocity gradient. Along these lines, it would also be helpful to clarify that in Eq. 1 for the geometry considered $\omega=0$.

Response: We thank the reviewer for this suggestion. We have updated the text and added Extended Data Fig. 13 accordingly.

7. Line 85-88: Will overexpression of RbmC and/or Bap1 to an even higher level diminish the reported pattern? Presumably with extremely strong surface adhesion, the cells will fail to verticalize and trigger the cascade of reorientation, right?

Response: We attempted to overexpress RbmC and Bap1 using our arabinose-inducible expression vector by increasing the amount of arabinose. At the highest arabinose concentration achievable in the experiment (2%), we still observe cell verticalization and radial alignment of cells. Beyond 2%, the metabolic burden of protein overexpression is too high and cells are unhealthy. We agree with the reviewer that suppression of cell verticalization is theoretically possible, however, the level of adhesion required is currently experimentally unfeasible.

8. Line 112-114: It's unclear which feature of Fig. 2c shows “a verticalized core that expands over time”. Specifically, the 15-19 h curves overlap with each other.

Response: The key feature distinguishing the verticalized core and the horizontal outer-rim is a downturn in the magnitude of n_z . This was not completely obvious in the original example in the manuscript, due to the large uncertainty in the measured n_{\perp} , which was due to limitations in this particular experiment in which we attempted to simultaneously track the mNeonGreen-labelled puncta and the mScarletI-labeled cells. To more accurately measure n_{\perp} , we now present data with mNeonGreen-labelled cells instead, which gives a much better signal-to-noise ratio and therefore allows for more accurate quantification of n_{\perp} . The updated results are given in Fig. 2c, in which one can more clearly see the expanding core.

9. Line 230-232: In experiments, the cells in the basal layer could also experience forces exerted by cells in the layer above. Could the authors comment on how the 3D structure might affect the mechanism of radial ordering, and more broadly the entire patterning dynamics?

Response: We thank the reviewer for this interesting question. Indeed, the confined biofilm also shows intriguing 3D organization, which is an active research direction in our team. We found that the basal layer organization, however, is not significantly affected by the 3D structure. This is because surface adhesion and frictional forces play a dominant role in setting the dynamics at the biofilm-glass interface. In fact, we find that the basal radial organization may actually modulate the 3D patterning dynamics that are observed in the bulk of the biofilm, but our data in 3D is still premature and we decide to focus on the basal organization in the present manuscript.

10. Fig.1a: The cells in the top surface of the biofilm also seem to possess a radial ordering. What is the origin of this ordering? Is it inherited from the order in the basal layer or does it arise via a distinct mechanism?

Response: We thank the reviewer for this keen observation. The physical environment of the cells at the top surface is very different because of the deformability of the gel. Our current hypothesis is that biofilm expansion introduces large tensile stresses in the gel, which are transmitted to cells at the top gel-biofilm interface. This stress transmission aligns the cells into a radial pattern on the top surface. This is reported in detail in our other work (see Zhang *et al. PNAS*. **118**, e2107107118. 2021). At the bottom interface, the solid substrate (glass) is not deformable, so the stress transmission mechanism does not apply.

11. Fig.1f: The $\Delta vpsL$ mutant doesn't form the same aster pattern as the WT* presumably because the adhesion proteins Bap1 and RbmC do not accumulate around the mutant cells (Berk et al. Science 2012). Is there an additional role that VPS plays in the reported self-patterning program? I.e., if $\Delta vpsL$ mutant cells were grown on surfaces coated with Bap1 and RbmC proteins, would they develop the same aster pattern as WT*?

Response: In the absence of VPS, Bap1 and RbmC are unable to bind cells to the surface because Bap1 and RbmC do not directly adhere cells to the surface; rather, they connect foreign surfaces to VPS, which in turn binds to the cells. In fact, the $\Delta vpsL$ mutant strain still produces these adhesins that localize on the glass surface, but due to the absence of VPS, cell-to-surface adhesion is not achieved (Yan *et al. PNAS*. **113**, e53375343. 2016). We performed the suggested experiment and consistent with the above-mentioned arguments, we did not observe the formation of aster patterns.

Reviewer 2:

This manuscript reports investigations on spatial structures of growing *Vibrio cholerae* biofilms. The authors used a combination of experimental, numerical and theoretical tools. They demonstrated a collective cell reorientation cascade in growing *Vibrio cholerae* biofilms that leads to a differentially ordered, spatiotemporally coupled core-rim structure. Cell verticalization in the core generates differential growth that drives radial alignment of the cells in the rim, while the radially aligned rim in turn generates compressive stresses that expand the verticalized core. Agent-based simulations and two-phase active nematic modeling reproduce the experimental observations and reveal the driving mechanisms underlying the differential ordering. In my opinion, the results in this well-written manuscript are novel and tell an interesting story. I recommend the publication of this manuscript after the authors address the following questions:

Response: We thank the reviewer for their support of our work. We address their comments point-by-point below.

1. All results in the manuscript are limited to colony size about 30 μm in radius. What limits the colony size? Would the radially aligned rim persist in colonies of much larger size.

Response: Beyond a certain size limit, the bacteria towards the center of the biofilm become nutrient limited and live imaging becomes impossible. We have attempted to image larger biofilms by replenishing the growth media every 24 hr and imaging after 72 hr. In addition, we use a DNA stain (SYTO 9) to label the cells, because the intrinsic fluorescence of the cells diminishes over time. An example biofilm grown for 72 hr is given in the figure on the right. Indeed, even for biofilms of diameter $\sim 140 \mu\text{m}$, the radial alignment persists. We believe this pattern would persist for all sizes of biofilms, because the self-reinforcing mechanism we identified in this manuscript should be valid for arbitrarily large biofilms. We have now included this example in Extended Data Fig 3.

A WT biofilm grown for 72 hr with media changes at 24 and 48 hr. Cells are stained with SYTO 9. Scale bar, 10 μm .*

2. Some data show imperfect radial alignment. For example, one can see some chirality in Extended Data Fig.2 d. Where does this weak chirality come from?

Response: Indeed, the cells do not always align into a perfect radial pattern, but rather, there is some randomness in their orientations, especially near the center of the biofilm. The randomness in the center is likely due to stochasticity in the initial conditions which dictate the initial pattern. When the biofilm is small and the number of verticalized cells is small, the driving force to radially align is weak. The imperfections due to the initial conditions and stochasticity in the growth and

division of the first few cells are frozen in place and can be observed in the mature biofilm. The randomness in the radial ordering in the rim, in the form of apparent “chirality”, is likely due to fluctuations in expansion, which could arise from noise in cell growth, substrate defects, local configuration variations, etc. Consider a local area that grows slower than its surroundings – cells in neighboring areas will grow into this area, resulting in local “bending” of the radial order. However, these fluctuations are local; we do not observe consistent, defined chirality that spans the entire biofilm.

3. What is the thickness of the agarose gel? Does the observed core-rim structure depend on the specific parameters of the agarose gel, such as thickness and elastic moduli?

Response: The estimated thickness of the gel is 500 μm , which is much larger than the height of the biofilm (at most around 20 μm). For gels with thickness on this scale, the thickness of the gel does not play an important role in the observed structure. Indeed, we have repeated the experiments for larger gel thicknesses and the overall core-rim structure was the same.

Furthermore, we find that for agarose gel concentrations in the range of 1.5-2%, there is no noticeable impact of the gel elastic modulus on the observed core-rim structure. In much softer gels, in which the confinement is weaker, we find that the biofilm still radially aligns, albeit to a lesser extent simply because the basal area is smaller (see our response to Reviewer 1, Q1 for our observations in unconfined biofilms). In the current paper, we choose to use relatively stiff gels (modulus ~ 60 kPa) so that the biofilm stays flat with a large basal area, allowing us to better investigate the mechanism driving radial organization.

Reviewer 3:

The authors present experimental results on growing colonies of *Vibrio cholerae* biofilms and show that, in the presence of adhesion between cells and the substrate, the basal layer of the colony forms a core of vertical cells at the centre and cells at the periphery orient radially and stay in the layer. The authors then explain this behaviour by considering a radial flow created by the verticalized core. Since the bacteria are elongated, they align to the direction of shear and as such bacteria at the rim align radially due to the ow created by the core. The radial alignment of the cells in the periphery generates compressive stresses that expand the verticalized core and, as a result, there is a feedback loop between the core and the rim which stabilises the pattern. To understand the behaviour observed in the experiment, the authors also show that the pattern can be captured by agent-based simulations of elongated rods which grow and divide and have surface adhesion with the substrate. In the absence of surface adhesion, neither orientation order nor a vertical core is observed and flows are random. This is correct for both experiments and agent-based simulations. Finally, a continuum model with two populations of in-plane and vertical cells is introduced to clarify the reason behind the formation of the pattern. In the continuum model, the in-plane and vertical populations transfer to one another due to a pressure. For the population with in-plane director, a 2D (in-plane) orientation field is considered. It is then shown that flows produced by inhomogeneous growth can lead to either aster or vortex orientations of the in-plane cells (depending on the core being a void growth region or a region with an excess growth).

The results and experimental observations are interesting, but there are a few points that need further clarification.

Response: We thank the reviewer for their enthusiasm towards our experimental results. We address their comments point-by-point below.

1. Since the adhesion with a substrate plays an essential role in the formation of the pattern, it should be clarified how and why the presence and absence of surface adhesion leads to different behaviours. In the SM, it has been argued that there is a “pinch-off” instability upon which the cells get ejected from the layer. It is discussed that, in the presence of the surface adhesion, the energy cost of this instability (for the values of the parameters in this experiment), is higher than the verticalization instability, and this energy difference vanishes in the absence of surface adhesion. This leads to spontaneous ejection of non-adhesive cells upon verticalization and as such, there is no vertical core in the non-adhesive mutant. In the paper in its current form, there is not enough evidence supporting the “pinch-off” instability. Considering the agent-based simulations and the experimental data, there is no measurement in the ejected cells. Is there any evidence in the experiments/simulations which support the ejection instability? In the experiment, if cells get ejected from the basal layer, the height of the colony should grow. As a result, comparing the height of the colony in the adhesive and non-adhesive case should show if such instability happens in the non-adhesive case or not. If it is not possible to judge from the experimental data about the height (due to any reason), it should be possible to see such an instability in the simulation of the rods (for example formation of local second layer in the simulations). Could authors provide such data to support the instability?

Response: We thank the reviewer for pointing out this weakness in our original manuscript. To accurately observe the ejection event from single-cell imaging requires full lineage tracking in 3D

biofilms, which is itself a formidable task. We do however find other alternative ways to show the pinch-off events. First, by examining closely the puncta trajectories in the nonadhesive ΔBC mutant, we see puncta disappear from the basal layer as a result of the cells being ejected into the third dimension. This ejection behavior is also evidenced in the kymographs below (Extended Fig. 11). In the WT* biofilm, puncta in the middle are arrested, whereas in the ΔBC biofilm, puncta trajectories abruptly end as a result of the corresponding cells being ejected from the basal plane.

Evidence for cell ejection. **a**, Time-lapse imaging of a growing puncta-labelled ΔBC biofilm showing the disappearance of puncta from basal layer over time. Scale bar, 10 μm . **b**, **c**, Kymograph of puncta trajectories taken from a rectangular cut through the center of the biofilm in a ΔBC mutant biofilm (**b**) and a WT* biofilm (**c**). **d**, Example trajectories of a single cell (lineage) from agent-based simulations without adhesion (top) and with adhesion (bottom).

In parallel, we probed the trajectory of cells in the agent-based simulations. We found that after the verticalization instability, characterized by a change in the orientation, cells without adhesion were more likely to be ejected from the surface, whereas cells with adhesion were more likely to remain verticalized and attached to the surface. We have added this and more quantification to the updated Extended Fig. 12.

We thank the reviewer for suggesting us to use the aspect ratio of the biofilm to indirectly test the ejection hypothesis. Unfortunately, looking only at the height of the biofilm is insufficient to validate the instability because both the WT* and ΔBC biofilms generate three-dimensional growth. We believe that the puncta-tracking experiment above is better suited to serve as the evidence for the pinch-off instability.

2. There has been a considerable amount of interest in orientation of bacteria in biofilms recently. Verticalization of the bacteria has been also observed in other experiments (for example Ref. 32).

The current manuscript lacks an overview and comparison between the experiments in this paper and other works that report formation of verticalized cores. As another example formation of a vertical core with radially aligned rim is also observed in "Spatiotemporal establishment of dense bacterial colonies growing on hard agar", however the focus of that paper is mainly on the role of nutrient depletion in a biofilm.

Response: We thank the reviewer for pointing out this past study. We have included the reference mentioned by the reviewer, along with other new citations. We have also now added a discussion to put our research into the context of these other works. Specifically, we note that although other works have discussed the verticalized core and the ejection of cells from the basal layer (Beroz *et al. Nat. Phys.* **14**, 954–960. 2018; You *et al. PRL.* **123**, 178001. 2019; Grant *et al. J. R. Soc. Interface* **11**, 20140400. 2014), it remained to be shown why adhesion facilitates *stably-anchored verticalized cells*, which we achieved in the current manuscript.

3. The authors consider the vertical core as an in-plane growth void. Does it mean that the vertical cells do not divide? Or do they divide but the mass increase goes to the third direction? Is it possible to do measurements on the growth to the third direction?

Azimuthally averaged, three-dimensionally resolved velocity field projected onto the $r - z$ plane.

Response: The reviewer is correct that verticalized cells are dividing and generating biomass in the third dimension, therefore not contributing to the growth in the basal layer anymore. Measuring the growth and flow pattern in *full* 3D is currently not possible with standard confocal microscopy due to photobleaching and phototoxicity. This technique can be performed using light-sheet microscopy (see Qin *et al. Science.* **369**, 71-77. 2020), which, unfortunately, is incompatible with our sample geometry. Upon the reviewer's request, we have further optimized our

puncta tracking experiments to quantify the 3D velocity field, *in the layers close to the substrate*. Preliminary data of the azimuthally averaged velocity field in the bottom $5 \mu\text{m}$ is given in the figure on the left. Indeed, the "growth void" in the center corresponds to a mass increase directed into the third dimension, consistent with the reviewer's picture.

4. The authors explain that the formation of the radial order in the rim is due to active flows and not due to passive interactions. It could be helpful to mention that the formation of nematic order due to active flows in a flow aligning regime has been studied before (See for example Sreejith Santhosh, et al, Journal of Statistical Physics,180 (2020)).

Response: We thank the reviewer for directing us to this reference; we found this reference and the references therein very helpful. We have now included a discussion about the formation of nematic order due to active flows in the text. One interesting distinguishing feature of our system is that "activity" in the form of growth is isotropic (in-plane), nevertheless, long-range order develops because the "activity" is spatially nonuniform.

5. In a similar basis, recent studies have shown that the growth into the third direction and verticalization can be caused by active "extensile" flows (such as flows created due to cell division). See for example "Extensile stress promotes out-of-plane flows in active layers, Mehrana Nejad, et al." and "Defect-mediated morphogenesis, Ludwig A. Hoffmann, et al". These may explain the formation of the vertical core in a dividing colony. Although the rim seems to be in the isotropic phase, the collision and the interaction between bacteria seem to be essential for the formation of the vertical core.

Response: We again thank the reviewer for bringing these works to our attention. In this manuscript, we have taken a different, complementary approach to model the out-of-plane flow, by considering mechanical instabilities at the single-cell scale. An alternative mesoscopic approach would be to consider quasi-two-dimensional or three-dimensional flows in an active layer. We choose the single-cell approach as it clearly distinguishes the anchoring dynamics in the WT* mutant from that in the adhesion-less mutant, whereas it is unclear how one would model the adhesion-driven anchoring in a continuum/mesoscopic theory.

6. Comparing Figs. 4b and 4d shows that the radial alignment parameter S is non-zero in the core, meaning that the core is not perfectly vertical but has a 3d splay form. This should be clarified and discussed in the main text.

Response: We thank the reviewer for this keen observation. Indeed, in the core of the biofilm, cells do not need to be completely vertical. We quantify the degree of verticalization as the fraction of cells which do not generate growth in the basal plane; this does not require $n_{\perp} = 1$. Instead, when n_{\perp} is larger than a threshold value of about 0.25 (depending on the length of the cell), the corresponding cells will send their offspring into the third dimension and effectively makes no contribution to the expansion in the basal layer. We now clarify this in the text. As a result, even "verticalized" cells have some radial component of their director and therefore non-zero S .

minor comments:

1. The definition of \hat{n}_{\parallel} in the radial order parameter S is not clear. Going through the SI it becomes clear, but it would be helpful to have it in the main text.

Response: We have now clarified the definition of \hat{n}_{\parallel} in the main text.

2. The data in Fig. 3d is widely spanned along y-axis. It could be helpful to show the average values (with error-bars) on top.

Response: We thank the reviewer for this helpful suggestion. We have now updated Fig. 3d to include superimposed error-bars to show the trend in the data more clearly.

3. Figs. 4 a)-d) show the pressure, fractional density of vertical cells, radial velocity, and radial order of cells as a function of position in the colony in different times. It could be helpful to scale the x -axis with the size of the colony at different times to make the comparison easier.

Response: We have included this rescaled data in Extended Data Fig. 20. We leave figure 4a-d as it is to facilitate comparisons with Fig. 2c-e.

4. In the line below Eq. 23 in the SM, ϵ needs to be larger than ϵ_0 for rod-shape particles.

Response: We thank the reviewer for pointing out this typo, this has been corrected in the SI.

Reviewers' Comments:

Reviewer #1:

Remarks to the Author:

The authors have satisfactorily addressed all of my concerns, and I am happy to recommend publication.

Reviewer #3:

Remarks to the Author:

The authors have successfully addressed my questions. I recommend the manuscript to be published in Nature Communications.

Reviewer #4:

Remarks to the Author:

The authors have now included new experimental data on puncta trajectories, to support the "pinch-off" instability introduced in the manuscript. They have also confirmed that surface adhesion in agent-based simulations keeps the vertical cells in the plane, whereas without surface adhesion cells get ejected from the basal plane. These have substantially clarified the important role of surface adhesion in the robust pattern observed in the experiment.

The authors have satisfactorily addressed all my concerns. I recommend publication.